# Factors associated with time to return to horse racing following a clavicle fracture in jockeys competing in Great Britain: A review and analysis of medical records

Laura J. Newton[1,2], Nick Dobbin[1], Peter Goodwin[1,3], Jennifer S. Crampton[1] *

1 Department of Health Professions, Faculty of Health and Education, Manchester Metropolitan University, Manchester, United Kingdom, 2 Cheshire and Wirral Partnership NHS Foundation Trust, Ellesmere Port Hospital, Ellesmere Port, Cheshire, United Kingdom, 3 Institute of Sport, Manchester Metropolitan University, Manchester, United Kingdom

* J.Crampton@mmu.ac.uk

## Abstract

### Purpose

Competitive horse racing is the second largest sport in Great Britain by spectator attendance, employability, and revenue. It is a lucrative yet hazardous sport, with high injury rates, particularly from falls. Clavicular fractures are one of the most common injuries reported, yet their management, especially regarding return to racing, is under-researched. The purpose of this study was to explore the factors associated with the time to return to competition following a clavicular fracture in jockeys competing in Great Britain.

### Materials and methods

This review of medical records utilised data from the British Horseracing Authority spanning 2011–2018, inclusive. Data included the jockey's age, sex, type of licence, race discipline, location of incident, and fracture management. Descriptive statistics and univariable and multivariable generalised linear models were constructed to analyse the impact of these factors on the time to return to racing.

### Results

Out of 212 records of clavicular fractures, 169 were analysed. The majority (82.8%) of fractures were managed conservatively, with the remainder requiring surgery. The median time to return to racing was 40 days, with an interquartile range of 34 days. Following a clavicular fracture, the results from the univariable and multivariable models indicated that the management approach, whether the fracture is displaced, and the type of race in which the injury occurred have the greatest influence on extending the time to return to racing. In contrast, professional, conditional and amateur licence types, as well as experiencing the injury at a racecourse, were associated with reduced time to return to racing, which may indicate greater risk-taking behaviour.

**Data Availability Statement:** The data used in this study is owned by the British Horseracing Authority and requested by the authors for the purpose of

this study. Verification of permission from the British Horseracing Authority was obtained prior ethics approval being granted. The data included various factors associated with time to return to racing as specified in the Methods along with a diagnosis and time to return. The data can be requested from the British Horseracing Authority at the following email address: info@britishhorseracing.com.

**Funding:** The author(s) received no specific funding for this work.

**Competing interests:** The authors have declared that no competing interest exist.

## Conclusion

This study offers unique insights into key medical and contextual factors that influence the time to return to racing among jockeys in Great Britain, contributing to tailoring medical management and return to racing protocols to support jockeys' health and career longevity. Clinicians working within horse racing can use the findings of this study to provide return to racing guidance to trainers, riders and other medical professionals based on the key contextual information reported in this study.

## Introduction

Horse racing is a physically demanding sport where riders compete in flat or jump racing. The sport is potentially lucrative for the riders (e.g., prize money, riding fee, sponsorship, media), racehorse owners (e.g., prize money), trainers (e.g., prize money), businesses (e.g., racecourses, hospitality, media), and the general public (e.g., gambling practices), hence its popularity [1] and contributing to an industry worth over £3.5 billion. In 2014, a report for the Department for Culture, Media and Sport [2] reported that owners, trainers, and jockeys received £98 million, £10 million, and £9 million, respectively, from the total prize money available (£123 million).

Whilst lucrative for many, the sport is recognised as being extremely hazardous regardless of competing at professional, amateur, conditional, or apprentice levels [3]. Several studies have reported a high injury incidence, many of which are traumatic due to falls from the horse [3–5], with approximately 50% of all falls being associated with injury. Current evidence indicates a considerably higher fall rate during jump racing (approximately 46 to 84 per 1000 rides) compared to flat racing (approximately 0.9 to 1.8 per 1000 rides) [5, 6]. However, it is important to consider the short distances and lack of obstacles in flat races, which results in a higher mean speed (greater than 18 m·s$^{-1}$) and reduced reaction time to apply any fall techniques when considering injury severity [4]. Furthermore, during flat races, there is a greater risk of a second impact caused by trampling due to the number of competitors [6].

Fractures are the second most common injury reported in flat (15%) and jump (18%) racing [7], with the shoulder girdle (shoulder/scapula/clavicle) being the most frequently injured region in jump racing. Clavicular fractures are also common [8] and result in a large number of compensatory claims made by riders (approximately £2000 per claim) [9]. Most clavicular fractures occur due to a direct impact to the shoulder girdle, with the middle third being the most frequent site of injury [10]. Despite being common, there has been little research into the management of clavicle fractures in jockeys with reference to the time to return to racing [11].

The treatment for a clavicular fracture depends on the location (middle third, lateral, medial) and type of fracture (undisplaced, displaced, comminuted). Traditionally, conservative management was favoured, involving a period of immobilisation with a simple arm sling, collar-and-cuff, or figure-of-eight brace in combination with analgesia and/or Kinesio tape [12, 13]. However, surgical fixation is recommended for displaced, shortened (greater than 20 mm), or comminuted midshaft clavicular fractures due to the high risk of non-union and poor patient-reported outcomes with conservative management [14]. Notwithstanding the complications that can occur with surgical management, such as neurovascular injury and infection, a recent study by Fahy et al. [11] reported that displaced midshaft fractures managed using open reduction internal fixation (ORIF) are effective and safe for professional jockeys in horse racing. It is reported that 95% of participants (*n* = 21) returned to competition at their pre-injury level with no complications reported with a mean time loss from competition of 68 days.

A study by Jack et al. [15] in professional American football players (*n* = 16) who experienced a clavicle fracture (no specific location identified) managed with ORIF reported that 94% of players returned to sport approximately 211 days later. In another study by Jack et al. [16], return rates for players managed conservatively (*n* = 32) had an average return to sport duration of 245 days, with an overall return to sport rate of 97%. Ahern et al. [17] found a difference in the time taken to return to sport between fractures managed conservatively compared with surgical management (61 ± 38 days vs. 100 ± 49 days; p = 0.008), with all participants returning to competition (*n* = 44). Regardless of treatment, there is a high level of return to sport following a clavicle fracture. However, it is clear that variability exists in the literature regarding the impact of the type of treatment used (conservative versus surgical) on the time taken to return to sport.

In addition to the management approach, there are other important considerations when assessing the time to return to racing following a clavicular fracture, such as the age of the rider, sex, race experience, location and type of incident, discipline returning to, and displacement of the fracture. Some of these risk factors are associated with fall risk more generally, regardless of injury type, but are important to consider when managing jockeys. For example, it has been reported that the length of career and riding experience are associated with a reduced risk of a fall [5, 7], whilst the discipline or type of race is associated with a greater fall risk [5]. These factors are likely to be important for medical personnel to consider when authorising a return to racing. It is also anticipated that the age of the rider, sex, access to medical support, and the injury itself (e.g., displacement) could be important to consider. However, these factors have yet to be modelled within statistical analyses despite potentially impacting the time to return to racing following a clavicular fracture.

Given the impact of clavicular fractures on finances, short-term performance goals within horse racing, and long-term health, it is important to understand the factors, including the management approach, that are associated with the time to return to competition whilst also ensuring that the long-term health of riders is prioritised. Therefore, this study sought to use an existing data source to answer the following research question: What factors are associated with the time to return to competition following a clavicular fracture in jockeys competing in Great Britain?

## Methods

### Study design

This study was a retrospective review of medical records (i.e., a chart review or clinical record review) controlled by the British Horseracing Authority (BHA). Permission to access this data was granted by the Chief Medical Advisor and approved by Manchester Metropolitan University (Application No. 10514). Written or verbal informed consent was not required as this study is a review and analysis of retrospective medical records owned by the BHA who acted as data controllers and had permission to share anonymous data for the purpose of research. This study followed the guidelines for conducting record review studies reported by Sarkar and Seshadri [18], namely identification of a data source (i.e., the BHA medical database), devising a data extraction strategy, extraction of the data, re-evaluation of a sub-sample of data, statistical analysis, and dissemination of the findings.

### Data retrieval process, inspection and analysis

**Retrieval.** The BHA medical database was used in this study. This database is managed by the Chief Medical Advisor and populated by physiotherapists and doctors on-site during races. Data retrieval took place on the 10th of June 2020 and included eight years of data, from

**Table 1. Dependent and independent variables requested from the BHA that were considered when developing the statistical models.**

| Unit | Factor | | Classification |
|---|---|---|---|
| Dependent variable | Time to return to racing | Continuous | Days |
| Independent variable | Sex at birth | Categorical | Male, female |
| | Licence | Categorical | Professional, amateur, conditional, apprentice |
| | Discipline | Categorical | Jump, flat, dual |
| | Location of incident | Categorical | Racecourse, other |
| | Incident type | Categorical | Race incident, other riding incident |
| | Race type where the incident occurred | Categorical | Steeplechase, Hurdle, Flat, Not race related |
| | Riding experience | Continuous | Years as a jockey |
| | Career races | Continuous | Total number of races completed |
| | Races per year | Continuous | Number of races per year |
| | Management approach | Categorical | Surgical, Conservative |
| | Classification of fracture | Categorical | Displaced, undisplaced, unknown |
| Excluded | Injured limb | Categorical | Right, left |
| | Location of fracture | Categorical | Mid, medial, lateral |
| | Weather conditions | Categorical | Dry and sunny, dry and cloudy, wet, mixed conditions |

1 January 2011 to 31 December 2018. This involved a formal request made by the research team with the relevant inclusion criteria included. The BHA then provided an encrypted version of the requested data in an anonymised form for further processing. A request was made for the data highlighted in Table 1, which reflected the primary outcome measure and independent factors included in the statistical analysis, as well as some factors later removed due to data insufficiency. The data reflected any incident that was reported to the BHA on race days, though the incident may have occurred prior to this. The research team were not directly involved in the extraction of data as this is kept confidential by the BHA. As such, data extraction using the criteria noted in Table 1 was conducted with the approval of the Chief Medical Advisor.

**Inspection.** Once received, the data was initially inspected by the principal researcher (LN) and subsequently categorised where appropriate. This data was then reviewed by all researchers to allow for critical debate regarding the relevance of the variables from a clinical standpoint with reference to the time to return to racing. Consequently, injured limb and weather conditions were removed as they were deemed not clinically relevant. The location of the fracture was deemed relevant but unfortunately contained substantial missing data (19/169, 89% missing). In the interest of rigour, this was removed as an independent variable.

**Data analysis.** Data was analysed using Microsoft Excel (Version 16.86, 2024, Microsoft Corporation, Washington, USA) and IBM SPSS Statistics (Version 29.0.2.0 for MacBook, IBM Corporation). Assumptions of normality were assessed through visual inspection of the quantile-quantile plot for all continuous variables (i.e., time to return to racing, riding experience in years, career races, number of races per year). All continuous variables failed to meet the assumption of normality due to positive skewness, hence median, interquartile range, minimum and maximum were reported. Categorical data was reported as frequency and proportions. Initially, a univariable model was constructed for each of the predictors using a gamma generalised linear model with a log link function applied and a deviance scale parameter method used. A multivariable model was then constructed using the same model structure with an 'enter' method used. Continuous factors were grand mean-centred and scaled before being included in each model. As the coefficients represented the change in the log of the mean of the dependent variable ($\ln(\mu)$), values were exported and exponentiated before being multiplied by the intercept to derive a point estimate and 95% compatibility intervals in raw

units (i.e., days). In addition, the percentage difference (categorical) or change from the intercept (continuous) was also estimated and the probability values (P value) presented in absolute terms to interpret compatibility. For practical context, time to return to racing was the dependent variable used in this study whereby an injury could equate to approximately 2–62 race days [6], 2–4 missed races per day [5], 11% of prize money, and risk of losing their riding position.

## Results

A total of 212 records of clavicular fractures were retrieved and transferred to the research team by the BHA. Of these, 6 (2.8%) were removed due to no return date being provided and 37 (17.5%) were removed due to concomitant injuries (e.g., rib fracture, concussion, spinal injury) that would likely influence the time to return to racing. This left 169 medical records that were included (male riders = 146 (86.4%), female riders = 23 (13.6%). For the records included, participants median age was 23.4 ± 5.7 (min = 17, max = 44) years with 5.0 ± 5.0 (min = 1.0, max = 27) years of riding experience. Riders had completed 315 ± 1146 (min = 1.0, max = 15039) rides in their career and 63.0 ± 133 (min = 1.0, max = 722) in the last year. Records included 75 amateur (44.3%), 45 conditional (26.6%), 38 professional (22.5%), and 11 apprentice (6.5%) riders who competed in the jump ($n$ = 118, 69.8%), flat ($n$ = 25, 14.8%), or dual ($n$ = 24, 14.2%) disciplines.

Of the 169 records included in the analysis, the vast majority of riding incidents that resulted in a clavicle fracture were from falls from the horse during a race ($n$ = 146, 86.9%), with the remainder occurring during a fall on the gallops ($n$ = 17, 10.1%) or in other riding incidents (e.g., injury in the stalls or a training yard incident; $n$ = 5, 2.9%). Accordingly, 88.1% ($n$ = 148) of all clavicle fractures occurred at the racecourse, with the remaining 11.9% ($n$ = 20) occurring at the gallops or trainer's yard. Of the 146 that occurred during a racing incident, 47.3% ($n$ = 69) occurred during a steeplechase race, 36.3% ($n$ = 53) occurred during a hurdle race and 16.4% ($n$ = 24) occurred on the flat.

Following a fracture, 140 riders (82.8%) were managed conservatively, while 29 riders (17.2%) opted for or required surgical management. Of the records available ($n$ = 40), 23 fractures (57.5%) were displaced, with the remaining 17 fractures (42.5%) being undisplaced; the status of all other fractures was unknown. Overall, a median time to return to racing of 40 ± 34 (min = 1, max = 262) days.

Results from the univariable and multivariable model are presented Table 2 and Fig 1. The values presented in Table 1 reflect the absolute number of days missed per factor derived from the log-linear coefficient that represents the change in the natural log of the mean of the dependent variable (S1 Table) and the exponentiated results (S2 Table). The univariable model indicated that licence type including amateur, conditional and professional resulted in 33 to 44% less time to return to racing compared to the apprentice riders (~78 days). Similarly, being a jump jockey resulted in 31% less time to return to racing compared to dual jockeys (~64 days), with little difference for a flat jockey (4%). Further, a fracture occurring in a hurdle race resulted in 16% less time to return to racing compared to a non-race related incident (~52 days). There was little influence of the location of the incident (-8%), the injury occurring in a steeplechase (-1%), an undisplaced fracture (10%) riding experience (-9%), career rides (-5%), and number of rides per year (-8%). When considering career races and number of races per year, a one standard deviation change (2371 career races and 139 races per year) resulted in a reduction in time to return to racing of 8 to 9% from the intercept (2 to 5 days). Clavicle fractures that occurred on the flat, that resulted in surgical management, and that were displaced increase the time to return to racing by 16 to 81%.

**Table 2. Overview of the univariable and multivariable models with data presented in days.**

| Independent variable | Univariable Model | | | | | Multivariable Model | | | | |
|---|---|---|---|---|---|---|---|---|---|---|
| | Point Estimate (days) | Lower 95% CI (days) | Upper 95% CI (days) | % | P value | Point Estimate (days) | Lower 95% CI (days) | Upper 95% CI (days) | % | P value |
| Sex | | | | | | | | | | |
| Male | 49.5 | 35.8 | 68.3 | -16 | 0.285 | 66.4 | 47.6 | 92.6 | 16 | 0.382 |
| Female (Ref) | **59.0** | - | - | - | - | **57.2** | - | - | - | - |
| Licence Type | | | | | | | | | | |
| Amateur | 52.3 | 33.0 | 82.9 | -33 | 0.090 | 41.4 | 20.5 | 84.0 | -28 | 0.371 |
| Conditional | 43.5 | 26.9 | 70.2 | -44 | 0.017 | 37.6 | 18.6 | 76.1 | -34 | 0.243 |
| Professional | 48.3 | 29.6 | 78.8 | -38 | 0.056 | 43.6 | 21.6 | 88.1 | -24 | 0.451 |
| Apprentice (Ref) | **77.9** | - | - | - | - | **57.2** | - | - | - | - |
| Jockey Type | | | | | | | | | | |
| Jump | 44.5 | 32.5 | 60.9 | -31 | 0.022 | 49.7 | 35.0 | 70.7 | -13 | 0.433 |
| Flat | 66.8 | 44.7 | 99.8 | 4 | 0.849 | 79.6 | 36.7 | 172.3 | 39 | 0.403 |
| Dual (Ref) | **64.3** | - | - | - | - | **57.2** | - | - | - | - |
| Location of Incident | | | | | | | | | | |
| Racecourse | 50.2 | 35.6 | 70.9 | -8 | 0.634 | 32.8 | 11.5 | 93.9 | -43 | 0.299 |
| Other (Ref) | **54.6** | - | - | - | - | **57.2** | - | - | - | - |
| Race Incident | | | | | | | | | | |
| Race Incident | 50.2 | 36.3 | 69.5 | -6 | 0.691 | 64.2 | 28.2 | 145.9 | 12 | 0.783 |
| Other Riding Incident (Ref) | **53.7** | - | - | - | - | **57.2** | - | - | - | - |
| Race Type | | | | | | | | | | |
| Steeplechase | 51.7 | 36.9 | 72.5 | -1 | 0.953 | 82.6 | 43.8 | 155.7 | 44 | 0.257 |
| Hurdle | 44.0 | 31.0 | 62.6 | -16 | 0.342 | 81.0 | 42.4 | 154.8 | 42 | 0.292 |
| Flat | 60.5 | 40.2 | 91.3 | 16 | 0.482 | 78.3 | 39.1 | 157.1 | 37 | 0.376 |
| Non-Race Related (Ref) | **52.2** | - | - | - | - | **57.2** | - | - | - | - |
| Management Approach | | | | | | | | | | |
| Surgical | 80.6 | 60.7 | 107.0 | 81 | <0.001 | 91.7 | 67.5 | 124.7 | 60 | 0.003 |
| Conservative (Ref) | **44.7** | - | - | - | - | **57.2** | - | - | - | - |
| Displacement | | | | | | | | | | |
| Displaced | 69.5 | 49.8 | 97.1 | 50 | 0.017 | 79.0 | 57.0 | 109.6 | 38 | 0.053 |
| Undisplaced | 51.1 | 34.5 | 75.6 | 10 | 0.627 | 66.8 | 46.2 | 96.4 | 17 | 0.411 |
| Unknown (Ref) | **46.3** | - | - | - | - | **57.2** | - | - | - | - |
| Riding Experience (years)* | 45.8 | 41.4 | 50.8 | -9 | 0.058 | 49.7 | 40.0 | 61.7 | -13 | 0.202 |
| Career Races (no.)* | 48.3 | 43.7 | 53.3 | -5 | 0.332 | 81.0 | 58.1 | 112.7 | 41 | 0.040 |
| Races per Year (no.)* | 46.2 | 41.7 | 51.3 | -8 | 0.097 | 37.9 | 28.2 | 51.1 | -34 | 0.007 |

Note: Data is presented in days derived from multiplying the exponentiated intercept with the exponentiated log-linear coefficient. % reflects the difference compared to the referent/reference group (categorical) or change from the intercept (continuous variables). * = grand mean centred and scaled. Ref = referent (reference) group. CI = 95% compatibility intervals. Bold = intercept.

In the multivariable model, licence type demonstrated a negative association for amateur, conditional and professional riders (~24 to 34% less time that apprentice) whilst surgical management was associated with greater time to return to racing (~60% more than conservative) (Table 2). However, this was not the case for sex, location of the incident, jockey type (flat only), race incident, and race type (all categories) that resulted in greater time to return to

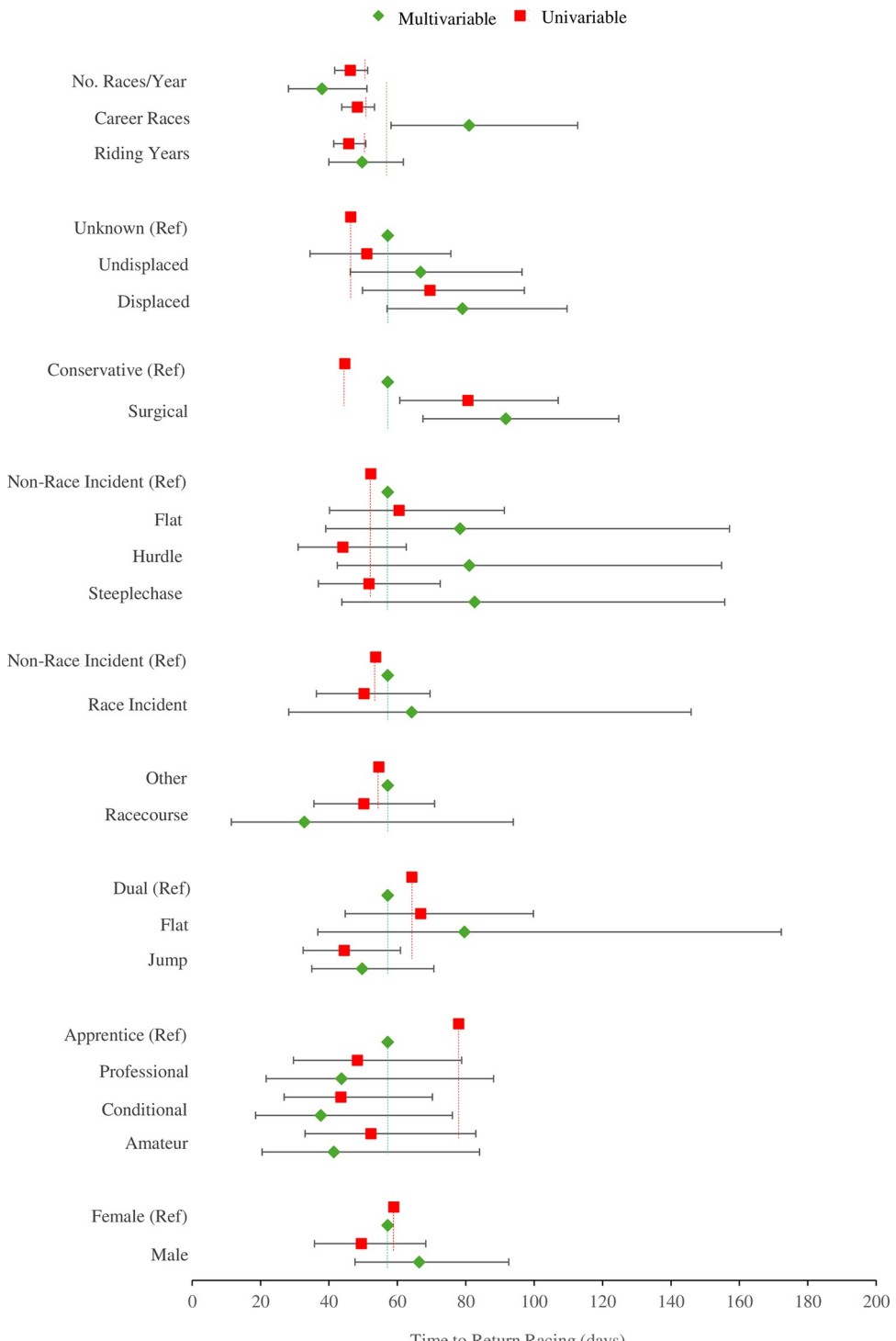

**Fig 1. Forrest plot that shows the association between each factor and time to return racing from a univariable (red squared) and multivariable (green diamonds) models.** Note: The red and green dashed lines reflect the intercept for the univariable and multivariable models, respectively.

racing when compared to the respective comparator group by 12 to 44% in the multivariable model. Career races also became positively associated with greater time to return to racing in the multivariable model such that each a one standard deviation change (4.8) resulted in a 41% increase equivalent to 24 days.

## Discussion

This study sought to understand the factors associated with time to return to racing among jockeys in Great Britain following a clavicle fracture caused by a riding incident. The review and analysis of medical records highlight several key findings. Firstly, the aetiology of clavicular fractures is best explained in this study as resulting from a fall from a horse at the racetrack. Secondly, regarding the clavicular fracture, we found that receiving surgical management, the displacement of the fracture, jockey type (flat only) and race type (flat) were associated with an increased time to return to racing compared to each respective referent (reference) group. Thirdly, licence type (all), jockey type (jump), riding experience, and the number of races per year were associated with reduced time to return to racing compared to their respective referent (reference) group or the intercept. This study offers unique insights into key medical and contextual factors that influence the time to return to racing amongst riders in Great Britain. These findings could be considered when deriving 'best practice' medical approaches, athlete management strategies, and return to racing policy and procedures.

Fractures reflect the largest number of injuries in horse racing, with athletes experiencing ~2.5 fractures during their career [8]. Of these, clavicle fractures represent the largest proportion of all fractures, carrying a 19% chance of occurring during a career [8]. In this study, a total of 212 clavicle fractures were recorded by the BHA over an 8-year period equating to an average of 27 being recorded each year. With the assumption that ~86% of all fractures involve the clavicle [8], this finding is similar to the average of 43 per year reported by O'Connor et al. [7] over a 4-year period and higher than average of 7 to 8 per year reported by Davies et al. [6] and Turner et al. [4] over an 11- and 8-year period, respectively. A total of 87 clavicle fractures were recorded as a result of a fall from the horse during a race, and this may potentially be linked to factors such as the racetrack surface, collision with other riders, and race obstacles [8]. A small proportion of jockeys experienced a clavicle fracture in the stalls, agreeing with previous work [8]. Also, in agreement with previous work (e.g., [6]), most fractures occurred during races that involve a jump such as a steeplechase, suggesting a much greater risk of a clavicle fracture compared to flat racing. Overall, the time to return to racing was 40 ± 34 days, with various factors explored in the study influencing this.

The effect of sex was inconsistent in this study demonstrating both negative and positive associations with time to return to racing. In both cases, the magnitude suggests a small effect despite some evidence suggesting that differences in sex hormones and inflammation can result in less efficient healing in females [19]. Total riding experience and the number of races per year were associated with less time to return to racing. Despite previous work indicating that younger populations have quicker fracture healing times [20] or that age does not affect clinical healing times for metacarpal fractures in adults [21], a negative association is important given the prevalence of smoking in jockeys [22], purposeful weight loss strategies [23], and the low osteogenic impact of horse racing [24] may impact the time to return, given the low bone mineral density pre- and post-injury that increases re-injury risk. The negative association may be due to greater levels of technical proficiency with increasing experience, which may result in a reduced risk of future falls [25]. Experienced jockeys may also be more familiar with managing injuries, having accumulated injuries over the years. This may also increase the risk of re-injury and impact the overall health and well-being of the jockey [26]. The total

number of races in a jockey's career was associated with an extended time to return to sport; this may be due to a higher incidence of injury as a result of greater exposure to the sport. We do note that the contrasting results for riding experience and career races in the multivariable model likely reflect the influence of other factors. For instance, some riders in the dataset had less experience but competed in more races (e.g., 5 years and 364 races *cf.* 6 years and 252 races). This may suggest that these factors are distinct and potentially the result of the level of success of trainers, the jockey's success, and attachment to specific training yards.

When compared to apprentices, riders with all other types of licences returned to racing, on average, 22 to 34 days sooner in the univariable model and 14 to 20 days in the multivariable model. This is similar for jockey type, with National Hunt jockeys returning sooner than flat and dual jockeys. A greater incidence of compromised bone health has been reported in flat jockeys compared to jump jockeys, which may impact bone healing explaining the 39% greater time [27]. Apprentice (flat) and conditional (jump) jockeys are younger and less experienced riders who receive a weight allowance for inexperience (known as a "claim"). Although there is a greater incidence of falls in National Hunt racing [5, 6], the injury severity is potentially greater in flat racing [4] due to a higher mean speed (greater than 18 m·s$^{-1}$) and reduced reaction time to apply any fall techniques. The high-energy trauma resulting from a fall in flat racing may impact the severity of the injury incurred, affecting the overall time to return to sport. Fractures that exhibit high-energy characteristics have been observed to be linked with an increased likelihood of non-union, which may be attributed to the severity of soft tissue damage, fragmentation of bone, or restricted blood flow to the location of the fracture [28].

Jockeys managed conservatively returned to sport sooner than those managed surgically. These findings align with the study by Ahearn et al. [17] (61 ± 38 days *vs.* 100 ± 49 days) but are not consistent with other studies where the average time to return to sport following fixation was 68 days [11], and with conservative management, it was 211 days [15]. Although all studies reported high levels of return to sport overall [11, 15, 17], complications such as wound infection, non-union, and neurovascular damage have been reported post-surgery [29], which may affect healing time and subsequent return to sport. However, a study by Fahy et al. [11] reported few post-operative complications, although the number of participants was relatively small [11]. Jockeys managed conservatively may be more likely to risk returning to sport sooner providing they can meet the BHA Medical Standards for Fitness to Ride [30]. These standards include achieving an appropriate pain-free range of movement, evidence of bone union shown radiographically, and clearance from their Orthopaedic Consultant and the Chief Medical Advisor. The accuracy of the data may also be a factor, as there may be a time-lapse between being physically fit to return to sport and the actual date inputted at the racecourse. If jockeys sustained the injury in the UK but are based in Ireland or France, the date of return to sport entered in the UK database may result in some degree of recall bias. The difference between return to sport for those with a known displaced fracture suggested greater time. There is evidence to support the surgical fixation of displaced, shortened, or comminuted midshaft fractures of the clavicle [14], although not all displaced fractures are managed with surgery. Therefore, it is difficult to draw definite conclusions about the effect of the type of fracture on return to racing in this study, though displaced fractures did appear to extend the time to return to racing in this study.

There were no notable differences in the time to return based on the type of race, though compared to non-race related, greater time was taken. The location of the incident (track or other e.g. gallops) or the incident type (in race or other riding incident (e.g. fall from a horse at the gallops) also had minimal impact on the time to return to racing. The reduced time between racecourse and other environments (e.g. gallops or trainer's yard) may be explained by the speed in which injuries are assessed. At a racecourse, an ambulance follows riders which

results in immediate medical assistance whereas outside a racecourse, jockeys will access medical services independently which may involve longer waiting times [31]. A study by O'Connor et al [26] found that only two in five jockeys reported an injury that occurred outside of the race immediately after an injury and just under a third never reported their injury to anyone. This delay in accessing treatment may influence the rehabilitation process and subsequent return to racing. Indeed, it's possible some did not report the injury until the next race when assessed pre-race. Finally, the incident type had minimal time to return to racing. This may be because the majority of the injuries reported in this study occurred because of the same mechanism of injury which was a fall from a horse as opposed to other incidents where injuries were sustained in the stalls or parade ring. Other studies have found similar findings whereby falls were the most common mechanism of injury [26, 32].

## Limitations

Whilst this study offers unique insights into some of the factors that might impact athletes' return-to-racing time, several limitations are worthy of note. One major limitation is the use of an existing dataset. Despite being populated by medical personnel and managed by the BHA, there is potential for the results to be biased towards more competitive incidents and subject to recall bias. Furthermore, the data captured only reflects a period up to 2018. We also note that incidents that occur on the gallops cannot be further defined as occurring at the trainer's yard or the preliminary gallop on the way to the starting gates. Finally, the factors included in the statistical analysis were limited by the availability and completeness of the data. Consequently, many other individual, medical, and contextual factors likely require further consideration for their impact on time to return to racing.

## Conclusions

The results of this study provide insights into the various individual, medical, and contextual factors that might impact the time to return to racing amongst jockeys in Great Britain. This study utilised a pre-existing data source controlled by the British Horseracing Authority (BHA), with injury data reported by trained medical personnel. Overall, our results indicate that the management approach, whether the fracture is displaced, and the type of race in which the injury occurred have an association with greater time to return to racing. In contrast, several factors were associated with reduced time, which may reflect greater risk-taking behaviour (e.g., licence type). These findings may inform future management approaches, return-to-racing protocols, and strategies to maximise long-term athlete health and well-being both during the injury period and post-return to racing. Clinicians working within horse racing can use the findings of this study to provide return to racing guidance to trainers, riders, and other medical professionals.

## Supporting information

**S1 Table. Log transformed results from the univariable and multivariable gamma generalised linear model.** Note: * = Grand mean centred and scaled. Ref = reference or referent group.
(DOCX)

**S2 Table. Exponentiated results from the univariable and multivariable gamma generalised linear model.** Note: * = Grand mean centred and scaled. Ref = reference or referent group.
(DOCX)

## Author Contributions

**Conceptualization:** Laura J. Newton, Peter Goodwin, Jennifer S. Crampton.

**Data curation:** Laura J. Newton.

**Formal analysis:** Nick Dobbin, Peter Goodwin, Jennifer S. Crampton.

**Investigation:** Laura J. Newton, Jennifer S. Crampton.

**Methodology:** Laura J. Newton, Nick Dobbin, Peter Goodwin, Jennifer S. Crampton.

**Project administration:** Laura J. Newton, Jennifer S. Crampton.

**Supervision:** Peter Goodwin, Jennifer S. Crampton.

**Visualization:** Nick Dobbin, Jennifer S. Crampton.

**Writing – original draft:** Laura J. Newton, Nick Dobbin, Peter Goodwin, Jennifer S. Crampton.

**Writing – review & editing:** Laura J. Newton, Nick Dobbin, Peter Goodwin, Jennifer S. Crampton.

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
