## [Decision Letter · Decision Letter 0]

9 Oct 2024

PONE-D-24-34824Factors associated with time to return to horse racing following a clavicle fracture in jockeys competing in Great Britain: A review and analysis of medical records.PLOS ONE

Dear Dr. Crampton,

Thank you for submitting your manuscript to PLOS ONE. After careful consideration, we feel that it has merit but does not fully meet PLOS ONE’s publication criteria as it currently stands. Therefore, we invite you to submit a revised version of the manuscript that addresses the points raised during the review process.

We look forward to receiving your revised manuscript.

Kind regards,

Chris Rogers

Academic Editor

PLOS ONE

Journal Requirements:

1. When submitting your revision, we need you to address these additional requirements. Please ensure that your manuscript meets PLOS ONE's style requirements, including those for file naming. The PLOS ONE style templates can be found at https://journals.plos.org/plosone/s/file?id=wjVg/PLOSOne_formatting_sample_main_body.pdf and https://journals.plos.org/plosone/s/file?id=ba62/PLOSOne_formatting_sample_title_authors_affiliations.pdf 2. Thank you for submitting your ethical approval document. We have noted that your document specifies that approval is granted for 6 months from the 30/10/2020. However, data was accessed in June 2020. Could you please submit the relevant ethical approval demonstrating that oversight was in place already in June 2020? 3. For studies involving third-party data, we encourage authors to share any data specific to their analyses that they can legally distribute. PLOS recognizes, however, that authors may be using third-party data they do not have the rights to share. When third-party data cannot be publicly shared, authors must provide all information necessary for interested researchers to apply to gain access to the data. (https://journals.plos.org/plosone/s/data-availability#loc-acceptable-data-access-restrictions)  For any third-party data that the authors cannot legally distribute, they should include the following information in their Data Availability Statement upon submission:1) A description of the data set and the third-party source2) If applicable, verification of permission to use the data set3) Confirmation of whether the authors received any special privileges in accessing the data that other researchers would not have4) All necessary contact information others would need to apply to gain access to the data. 4. Please include captions for your Supporting Information files at the end of your manuscript, and update any in-text citations to match accordingly. Please see our Supporting Information guidelines for more information: http://journals.plos.org/plosone/s/supporting-information.

Additional Editor Comments:

Thank you for you patience with the review process. Both reviewers had only minor edits for the manuscript.

Reviewers' comments:

Reviewer's Responses to Questions

**Comments to the Author**

1. Is the manuscript technically sound, and do the data support the conclusions?

Reviewer #1: Partly

Reviewer #2: Yes

2. Has the statistical analysis been performed appropriately and rigorously? 

Reviewer #1: Yes

Reviewer #2: Yes

3. Have the authors made all data underlying the findings in their manuscript fully available?

Reviewer #1: No

Reviewer #2: Yes

4. Is the manuscript presented in an intelligible fashion and written in standard English?

Reviewer #1: Yes

Reviewer #2: Yes

5. Review Comments to the Author

Reviewer #1: PONE-D-24-34824

Factors associated with time to return to horse racing following a clavicle fracture in jockeys competing in Great Britain: A review and analysis of medical records

General

This study reviewed records of clavicle fracture in jockeys and conducted a simple regression analysis of time to return to racing. Specific comments and suggestions to improve the manuscript are below.

Ethics Statement

Suggest addition - “Written or verbal informed consent was not required as this study is a review and analysis of retrospective medical records”.

Data availability

Expand on “some restrictions will apply”. Sometimes a minimal, summarised and de-identified dataset can be included and/or uploaded with restrictions to Figshare or similar.

Abstract

Line 57 – specify jockey age and sex, to differentiate from horse.

Line 58 and throughout – replace “accident” with “incident”.

Line 59 – univariable or multivariable models?

Line 61 – were all the records of clavicular fractures?

Line 62 – The standard deviation is high compared to the mean, consider checking normality and reporting median and IQR if skewed.

Line 63 – suggest “Following fracture, jockeys with more career race rides, but fewer rides on average per year, took longer to return to racing.”

Line 66 – Does minimal impact mean not statistically significant?

Line 69 – This is just a repeat of the results. What are the implications?

Results – should include coefficients and 95% Cis in abstract.

Introduction

Line 75 - While this history is interesting I don’t see the relevance to this study. The entire first paragraph is of little relevance, and could be condensed to one or two sentences that focuses on the jockey or deleted entirely.

Line 79 – delete “both”. Not sure of need of bracketed info.

Line 82 and 85 – Essentially a repeat of previous sentence. Could condense.

Line 103 – per claim?

Line 120 to 130 – New paragraph. The comparison to other sports is interesting.

Methods

Line 181 – provide initials of researcher in brackets.

Line 182 – What constituted too much missing data? Normally >10 or 20%.

Line 183 – Replace “removed” with “excluded”.

Line 187 – How much was missing? Report as xx/xx (xx%).

Line 188 – It is fine to include the description of these even with some missing data, it will just have a different denominator. It is only necessary to exclude during multivariable analysis.

Line 195 – Was normality of the data checked to ascertain whether mean and sd were appropriate descriptors?

Line 202 – As the data was provided de-identified, did you have access to the individual jockey % wins and places, and riders per season? Or were all based on the average?

Line 205 – Was p<0.05 the level of statistical significance?

Results

Line 207 – All clavicular fractures?

Line 208 – report total removed as percentage 43/212 (20.3%)

Line 211 – Include percentages for jockey sex.

Line 211 to 213 – Suggest including min- max also.

Line 217 – “Of the 169 records included in analysis…”

Line 232 to 245 and Figure 1 – This paragraph is nice, but the full results of the model need to be presented in a Table with coefficients, 95% CI and p-values and it would be helpful to include columns for the raw descriptive data in this table also. These look to be reported in the Figure but the resolution is poor and cannot be read sufficiently.

For the binary and categorical variables it is also unclear which is the reference category.

Racecourse appears to have been entered in the model as a continuous variable.

Were interactions explored? Riding experience is likely correlated with licence and age.

Normalisation to 90 days doesn’t make much sense as it appears time to return to racing can be a minus value.

Unclear whether univariable or multivariable analysis was attempted.

Discussion

Line 250 – As other fractures in the same time period were not reported, how can it be ascertained that clavicle fractures were common?

Line 253 and 256 and 287 and abstract– greater number of rides and greater riding experience are contradictory findings. Please check directions of association. What is the difference between total riding experience and total number of race rides in their career? Perhaps needs clarification if these are distinct.

Line 270 – typo or clunky sentence.

Conclusion – This paragraph is largely a repeat of the results rather than the implications or need for future studies. Is it possible to be more specific about how this information informs management decisions or policy or protocols etc?

Tables – Headings should be standalone.

Table 1 – replace “scale” with “continuous”.

References – Incorrect formatting of websites and reports. Missing dates/years, publisher, country etc.

Reviewer #2: Only a couple of minor comments.

Line 103 compensatory claims – assume figure is per unit claim – maybe add per claim within brackets

Results

Lines 218-219 – based on the data a bit of confusion which may be just around language used, or my reading in haste,

You mention with the remainder occurring during a fall on the gallops – do you mean when at the trainers yard and out training the horses on the gallops, or the preliminary gallop, on the way to the starting gates. – it may be useful to provide a case definition somewhere in the materials and methods eg all clavicle records on file with the BHA or all clavicle fractures occurring on raceday

6. PLOS authors have the option to publish the peer review history of their article (what does this mean?). If published, this will include your full peer review and any attached files.

Reviewer #1: No

Reviewer #2: No

---

## [Author Response · Author response to Decision Letter 0]

23 Nov 2024

Dear Professor Rogers, 

Thank you for handling this paper and processing this in a timely fashion. We have provided a response to each minor point noted by the reviewers. We hope this is deemed acceptable by yourself and the reviewers, 

Best wishes, 

Jenny Crampton 

2. Thank you for submitting your ethical approval document. We have noted that your document specifies that approval is granted for 6 months from the 30/10/2020. However, data was accessed in June 2020. Could you please submit the relevant ethical approval demonstrating that oversight was in place already in June 2020?

Thank you for raising this. The date noted above is the end date of the project, meaning that ethics remains in place at MMU 6 months from this date. The approval date, as specified in the letter, was the 8th of July 2019. Therefore, technically speaking, approval was in place from the 8th of July 2019 until 30th April 2021. 

3. For studies involving third-party data, we encourage authors to share any data specific to their analyses that they can legally distribute. PLOS recognizes, however, that authors may be using third-party data they do not have the rights to share. When third-party data cannot be publicly shared, authors must provide all information necessary for interested researchers to apply to gain access to the data. (https://journals.plos.org/plosone/s/data-availability#loc-acceptable-data-access-restrictions) 

4) All necessary contact information others would need to apply to gain access to the data.

We do not have the permissions in place to make the data publicly available, so we have provided a statement that covers the four points above in the submission form: “The data used in this study is owned by the British Horseracing Authority and requested by the authors for the purpose of this study. Verification of permission from the British Horseracing Authority was obtained prior ethics approval being granted. The data included various factors associated with time to return to racing as specified in the Methods along with a diagnosis and time to return. The data can be requested from the British Horseracing Authority at the following email address: info@britishhorseracing.com”. 

4. Please include captions for your Supporting Information files at the end of your manuscript, and update any in-text citations to match accordingly. Please see our Supporting Information guidelines for more information:http://journals.plos.org/plosone/s/supporting-information.

No caption is required as there is no supporting information. 

Done

Additional Editor Comments:

Thank you for your patience with the review process. Both reviewers had only minor edits for the manuscript.

Thank you. 

Reviewers' comments:

Reviewer's Responses to Questions

Reviewer #1: PONE-D-24-34824

General

This study reviewed records of clavicle fracture in jockeys and conducted a simple regression analysis of time to return to racing. Specific comments and suggestions to improve the manuscript are below.

We thank the reviewer for taking the time to review our manuscript and providing constructive feedback that we feel has enhanced the work substantially. We appreciate that this is unpaid work, so thank the reviewer for supporting our work and the publishing process. 

Ethics Statement

Suggest addition - “Written or verbal informed consent was not required as this study is a review and analysis of retrospective medical records”.

Thank you for the suggestion. This has been included in the ethics statement as well as in the Methods (line 157). 

Data availability

Expand on “some restrictions will apply”. Sometimes a minimal, summarised and de-identified dataset can be included and/or uploaded with restrictions to Figshare or similar.

We thank you for the suggestion. This was noted by the editor, and we have provided a data availability statement within the submission.

Abstract

Line 57 – specify jockey age and sex, to differentiate from horse.

This has now been included. 

Line 58 and throughout – replace “accident” with “incident”.

Done

Line 61 – were all the records of clavicular fractures?

Yes, this has been clarified. 

Line 63 – suggest “Following fracture, jockeys with more career race rides, but fewer rides on average per year, took longer to return to racing.”

Thank you, this has been incorporated. 

Line 66 – Does minimal impact mean not statistically significant?

We understand the point being made here; however, we are reluctant to base our interpretation of the results solely based on P values and “significance”. As noted in several statistics guidelines, this should be avoided where possible. We feel that minimal is a suitable quantification of an effect size which this case in the estimated change in the intercept. We hope you understand our rationale. 

Line 69 – This is just a repeat of the results. What are the implications?

Thank you, this has now been improved. 

Results – should include coefficients and 95% Cis in abstract.

We appreciate the point here, but given the approach to the statistics, we feel this might be overwhelming in an Abstract and take it beyond the word count. We have now provided an overview using a combination of days and percentage difference/change that better reflect the results. 

Introduction

Line 75 - While this history is interesting, I don’t see the relevance to this study. The entire first paragraph is of little relevance and could be condensed to one or two sentences that focuses on the jockey or deleted entirely.

We can see the point being made here, particularly with reference to the first two sentences. These have now been removed, but we are keen to keep the context around finances as we consider this important when discuss time lost during the injury. 

Line 79 – delete “both”. Not sure of need of bracketed info.

This has been deleted. 

Line 82 and 85 – Essentially a repeat of previous sentence. Could condense.

We have removed aspects here but have kept the final sentence. 

Line 103 – per claim?

This has been added. 

Line 120 to 130 – New paragraph. The comparison to other sports is interesting.

Thank you, we have now separated this section as suggested. 

Methods

Line 181 – provide initials of researcher in brackets.

Added

Line 182 – What constituted too much missing data? Normally >10 or 20%.

This has now been clarified as > 3 independent variables. 

Line 183 – Replace “removed” with “excluded”.

Corrected. 

Line 187 – How much was missing? Report as xx/xx (xx%).

This has been added.

Line 188 – It is fine to include the description of these even with some missing data, it will just have a different denominator. It is only necessary to exclude during multivariable analysis.

We accept the point here regarding the multivariable analysis, but with 89% of the data missing we feel that the low statistical power and type II error rates here would detract from the work. We, therefore, feel it’s best to remove this data but keep the statement on Line 188. 

Line 202 – As the data was provided de-identified, did you have access to the individual jockey % wins and places, and riders per season? Or were all based on the average?

We have provided the number of rides per season as we had this per rider, but we are unable to comment on the %wins and places. 

Line 205 – Was p<0.05 the level of statistical significance?

We have opted to report the P values as they appear rather suggesting an arbitrary cut-off value. We feel it’s essential for the reader to decide their own level of a type II error rather than suggesting 0.05. We do note that this aligns with the recommendations of the American Statistical Society other seminal papers. 

Results

Line 207 – All clavicular fractures?

Yes, this has been clarified. 

Line 208 – report total removed as percentage 43/212 (20.3%)

We have now included the percentage in brackets. 

Line 211 – Include percentages for jockey sex.

This has been included. 

Line 217 – “Of the 169 records included in analysis…”

Thanks, done. 

Line 59 – univariable or multivariable models?

Line 62 – The standard deviation is high compared to the mean, consider checking normality and reporting median and IQR if skewed.

Line 195 – Was normality of the data checked to ascertain whether mean and sd were appropriate descriptors?

Line 211 to 213 – Suggest including min- max also.

Line 232 to 245 and Figure 1 – This paragraph is nice, but the full results of the model need to be presented in a Table with coefficients, 95% CI and p-values and it would be helpful to include columns for the raw descriptive data in this table also. These look to be reported in the Figure but the resolution is poor and cannot be read sufficiently.

For the binary and categorical variables it is also unclear which is the reference category. Racecourse appears to have been entered in the model as a continuous variable. Were interactions explored? Riding experience is likely correlated with licence and age. Normalisation to 90 days doesn’t make much sense as it appears time to return to racing can be a minus value. Unclear whether univariable or multivariable analysis was attempted.

We hope you don’t mind us grouping these points, but we felt it was easier to address them all together given the changes we have made in the statistics and results. 

Firstly, we would like to acknowledge that we re-checked the distribution, and whilst it appears fairly normal, visually based on the Q-Q plot, there is a degree of positive skewness that we had previously overlooked. As such, we have re-analysed the full dataset. Continuous variables are now presented as the median and interquartile range, and where appropriate, we have also provided the minimum and maximum values.

Given the positive skewness, we felt it was appropriate to derive new models as these were originally based on the assumption of a Gaussian distribution.

To address the issue of positive skewness, we have now applied a gamma generalised linear model with a log link function and a deviance scale. This approach has provided a much stronger model fit and better represents the dataset. You mentioned the use of univariable and multivariable models. Initially, we had only conducted a multivariable model using the enter method. However, we recognise the value of including both univariable and multivariable models, and therefore, both are now included.

In using the gamma model with a log link function, we felt the readability and usability of the work were somewhat reduced. To address this, we exponentiated the coefficients, which reflect the change in the log of the mean of the dependent variable and have included both the original and exponentiated models in the supplementary material. We then used the exponentiated values and multiplied them by the exponentiated intercept to derive values in days. This data has been incorporated into the manuscript to enhance readability, rather than presenting log-transformed or exponentiated log data. The full model, reflecting values in days, is now presented in a table that includes the point estimate, the upper and lower compatibility/confidence intervals, the percentage difference or change, and the p-values. A figure is also included to illustrate the change/difference in days from both the univariable and multivariable models with the individual or pooled intercept displayed. 

To respond fully to the above points, we have not interacted the data but have removed age from the model, given the high degree of shared covariance. We reviewed location and confirm that it is categorical. The table and figure now indicate the reference/referent groups for ease of understanding.

Thank you for prompting us to reflect on the original analysis. Whilst it was potentially appropriate, we feel this re-analysis provides a much more comprehensive overview with reduced overdispersion. Reassuringly, our results were not substantially different to the original submission, but we are more confident in the current approach.

Discussion

Line 250 – As other fractures in the same time period were not reported, how can it be ascertained that clavicle fractures were common?

This has now been removed. 

Line 253 and 256 and 287 and abstract– greater number of rides and greater riding experience are contradictory findings. Please check directions of association. What is the difference between total riding experience and total number of race rides in their career? Perhaps needs clarification if these are distinct.

We have now re-analysed the results. At a univariable level, riding experience and number of rides do suggest similar associations; that is, they are both associated with slightly reduced time to return to racing. In the multivariable model, these associations are opposite to each other as originally stated with career rides (greater number of rides) being associated with greater time to return. We do appreciate the point raised, but we do feel these reflect distinct points. The number of years of being a jockey does not necessarily mean a greater number of rides whilst there may be some who are highly successful and race frequently despite being younger than others. For example, in our dataset, we have an individual with 21 years’ experience who has completed 6307 races whereas another has 18 years of experience and completed 11352 races. At the lower end, we have an individual with 5 years’ experience and 364 races cf. 6 years and 252 races. 

We have sought to provide further clarification on this point in the manuscript but have not made substantial changes given the association in the multivariable model supports our previous findings. 

Line 270 – typo or clunky sentence.

This sentence has been amended. 

Conclusion – This paragraph is largely a repeat of the results rather than the implications or need for future studies. Is it possible to be more specific about how this information informs management decisions or policy or protocols etc?

Thank you for highlighting this. We have now made substantial revisions to the conclusion to offer a more informative overview with consideration for the implications this might have for those working in horseracing (e.g., those making decisions in return to racing). 

Tables – Headings should be standalone.

We have provided the Tables, Heading and Notes at the end of the document now as per 

Table 1 – replace “scale” with “continuous”.

Done

References – Incorrect formatting of websites and reports. Missing dates/years, publisher, country etc.

Apologies, reference 2, 3 and 20, 31 and 32 have now been correct (ref 1 was removed). 

Reviewer #2: Only a couple of minor comments.

We thank Reviewer 2 for their time and effort reviewing our manuscript. We appreciate that they have taken the

---

## [Editor Report · Decision Letter 1]

6 Jan 2025

Factors associated with time to return to horse racing following a clavicle fracture in jockeys competing in Great Britain: A review and analysis of medical records.

PONE-D-24-34824R1

Dear Dr. Crampton,

We’re pleased to inform you that your manuscript has been judged scientifically suitable for publication and will be formally accepted for publication once it meets all outstanding technical requirements.

Kind regards,

Chris Rogers

Academic Editor

PLOS ONE

Additional Editor Comments (optional):

Thank you for the thorough and detailed changes made in response to the reviewers comments.
---

## [Editor Report · Acceptance letter]

10 Jan 2025

PONE-D-24-34824R1 

PLOS ONE

Dear Dr. Crampton, 

I'm pleased to inform you that your manuscript has been deemed suitable for publication in PLOS ONE. Congratulations! Your manuscript is now being handed over to our production team.

Kind regards, 

on behalf of

Dr. Chris Rogers 

Academic Editor

PLOS ONE
